# Impact of Hotel Employees' Psychological Well-Being on Job Satisfaction and Pro-Social Service Behavior: Moderating Effect of Work–Life Balance

**Hyo-Sun Jung [1][ID], Yu-Hyun Hwang [2] and Hye-Hyun Yoon [2],***

[1] Center of Converging Humanities, KyungHee University, Seoul 02447, Republic of Korea; chefcook@khu.ac.kr
[2] College of Hotel & Tourism Management, KyungHee University, Seoul 02447, Republic of Korea; yuhyunhwang@naver.com
* Correspondence: hhyun@khu.ac.kr

**Abstract:** This study investigates how deluxe hotel employees' perceptions of their own psychological well-being impact their job satisfaction and pro-social service behavior. It also examines the moderating effect of work–life balance on the relationship between psychological well-being and job satisfaction. A self-administered questionnaire was distributed to 275 deluxe hotel employees using convenience sampling. First, of the studied sub-factors of employee psychological well-being, positive relationships increased job satisfaction the most, followed by self-acceptance, purpose in life, and environmental mastery. Second, deluxe hotel employees' job satisfaction positively impacted their pro-social service behavior. Third, the positive effect of one sub-factor of psychological well-being, purpose in life, had a stronger impact on job satisfaction in respondents with high levels of work–life balance. Theoretical and practical implications, as well as limitations and future research directions, are discussed.

**Keywords:** psychological well-being; job satisfaction; work–life balance; pro-social service behavior; deluxe hotel employee

## 1. Introduction

COVID-19 has had a devastating effect on the hospitality industry, including hotels around the world [1–5]. Up to this point, studies linking the hospitality industry and COVID-19 have received a lot of academic attention [6]. In particular, the majority of studies have focused on the negative impact of COVID-19 on members of organizations [7]. In particular, deluxe hotels are a type of business that has been directly hit by travel restrictions and decreased travelers due to COVID-19, which has resulted in multiple cases of hotel closures, downsizing, and layoffs [8]. The severity and uncertainty of COVID-19 effects are closely related to the psychological well-being of organizational workers [9,10]. A number of studies have demonstrated the positive effect of employees' psychological well-being on their performance [11,12]. Furthermore, today, many talented people are quitting their jobs because of psychological problems caused by depression and burnout in the workplace [13]. According to a previous study of the hospitality industry, employees who are creative, perform well, and have high levels of participation generally experience well-being at work [13]. Fredrickson [14] pointed out that, since a positive feeling of well-being enables sustained development of personal resources, happy people continue to make positive contributions to an organization. Haller et al. [11] reported that the frequency of pro-social behavior is consistently higher among people with high levels of well-being, which implied that those with high well-being are more likely to participate in voluntary actions. Dakin et al. [15] stated that psychological well-being is closely related to happiness and observed that employee happiness promotes customer-oriented behavior.

An individual's ability to respond to difficulties in the workplace can also be impacted by their level of psychological well-being, which in turn affects job satisfaction [16].

Although a number of studies have demonstrated that workers who are happy and have high levels of psychological well-being tend to be more productive than their less happy counterparts [17], virtually no study has explored the link between well-being and pro-social service behavior. In particular, studies conducted on deluxe hotel employees are very limited, and there are almost no studies that examine the organic, causal relations between the sub-factors of psychological well-being, job satisfaction, and pro-social service behavior. In addition, most of the studies that have already been conducted have either used psychological well-being as a final dependent variable [18] or have examined simple causal relationships [19]. Work–life balance, which is a moderating variable in the current study, is recognized as a key element in human resources and has been proved to positively impact organizational performance regarding employee job satisfaction or career success [20,21]. From an employer's perspective, helping employees achieve good work–life balance provides a competitive advantage that enables organizations to recruit and retain qualified workers [22]. In contrast, poor work–life balance can be viewed as a work-related stress factor and can have a negative impact on an organization [23]. This can also be explained by social cognitive theory, which states that an individual's behavior is not created by internal forces or external stimuli but that cognition, behavior, and environmental factors are interconnected [24].

This study investigates how deluxe hotel employees' perceptions of their own psychological well-being impact their job satisfaction and pro-social service behavior. In particular, this study intends to explore the relative influences of sub-variables of psychological well-being on job satisfaction. It also examines the moderating effect of work–life balance on the relationship between psychological well-being and job satisfaction.

## 2. Literature Review and Conceptual Model

### 2.1. Psychological Well-Being in the Hospitality Industry

Multiple studies have found that psychological well-being is one of the most important factors impacting an organization's success [25–27]. Therefore, in the current analysis, psychological well-being is used as a major independent variable. In particular, the psychological well-being of hotel employees is even more important because they interact with customers at the forefront of service, thereby producing results [28,29]. Several recent studies have examined psychological well-being among workers in the hospitality industry. Zulkarnain and Akbar [30] reported that hotel employees' turnover intentions reduced when they felt that their needs were met by an organization's efforts to improve employee psychological well-being. Agarwal [19] found that psychological well-being was affected by personal and circumstantial factors and concluded that consistent communication, positive relationships, and increased autonomy increased hotel employees' psychological well-being. Bayighomog and Arasli [13] found that spiritual well-being significantly and positively impacted hotel employees' customer-focused boundary-spanning behaviors. Subramony et al. [31] argued that frontline employees' well-being could be enhanced when they experienced fewer tense processes (anxiety) in their work lives and when they felt supported by their organization. According to Kim and Jang [32], because hospitality industry workers suffer from diverse energy-related symptoms and frequently have no opportunities to rest and recharge, improvements to the physical environment alone, such as staff break rooms, could enhance their psychological well-being.

### 2.2. Psychological Well-Being, Job Satisfaction, Pro-Social Service Behavior, and Work–Life Balance

Different scholars offer different definitions of well-being, and no single definition has been established [33]. In the existing research, well-being is broadly divided into subjective well-being and psychological well-being. While the concept of subjective well-being falls within hedonic perspectives, psychological well-being is understood from the perspective

of self-fulfillment [33]. Subjective well-being, subjective quality of life, life satisfaction, and happiness are similar, closely related concepts [34]. They are measured using both negative aspects—such as depression, feelings of helplessness, and diminished self-esteem—and positive aspects, such as high self-esteem and general life satisfaction. The current study focuses on psychological well-being. The concept of psychological well-being was first seriously examined by Ryff [35], who argued that quality of life should be understood based on how well a person functions as a member of a society, rather than only on individual happiness or life satisfaction. Psychological well-being is based on the true realization of an individual's potential and their pursuit of perfection. Although it is not directly related to happiness, it can be considered as a byproduct of a good life [36]. Ryff [35] devised a scale of psychological well-being that measures individual psychological well-being based on psychological theories. This scale addresses the following six subfactors: (1) autonomy, or alignment with one's personal beliefs; (2) personal growth, or realizing one's talent and potential; (3) environmental mastery, or managing one's own life; (4) positive relationships, or forming close bonds with others; (5) self-acceptance, or recognizing one's personal limits and knowing and accepting oneself; and (6) purpose, or experiencing a sense of meaning, purpose, and direction in one's life.

Locke [37] described job satisfaction as an employee's positive emotions toward their job and defined it as a pleasant, positive condition that results from positive evaluations of an individual's work or working experience. According to Porter and Steers [38], job satisfaction meant that people believe that the real results of their work justify their efforts and noted that negative emotions about work increase when these results are not satisfactory. Szilargy and Wallace [39] defined job satisfaction as a person's attitude toward their job itself, as well as toward job-related matters, such as rewards and supervision; they asserted that job satisfaction includes cognition, emotion, and behavioral inclination. Brief and Weiss [40] defined job satisfaction as an internal state that is revealed by the emotions people experience when they cognitively or emotionally assess their work. According to Arnold and Feldman [41], job satisfaction describes mindset, emotions, and feelings toward one's work that are generally positive. Luthans and Sommer [42] defined job satisfaction as a positive emotional reaction to a work situation and found that it is determined by how much rewards approach or exceed an employee's expectations.

Pro-social service behavior means voluntary behavior by point-of-service employees that aims to promote an organization's success by providing higher-quality service to external customers [43,44]. It refers to employee behaviors that directly help an organization or other organizational members. It can be defined as employees' direct actions that promote personal or organizational benefits [45,46]. The research on pro-social service behavior originated in studies of pro-social organizational behavior. Although some scholars have offered slightly differing definitions, pro-social service behavior is generally defined as behavior that benefits others, regardless of the motivation behind it [47]. The terms pro-social behavior, pro-social service behavior, and service-providing behavior are often used interchangeably. Pro-social service behavior, which contrasts with anti-social behavior, such as selfishness and lack of trust within an organization, refers to actions aimed at helping others without the expectation of an external reward. They are actions that are voluntarily performed to benefit others and an organization [48].

Work–life balance refers to an individual's subjective perception that their work and personal life are aligned with their values and preferences [49]. Research on work–life balance is common in the fields of human resources management, organizational psychology, and home economics. The topic has received particular attention in the field of human resources development because work–life balance is closely related not only to employees' individual performances and the performance of an organization as a whole, but also to variables impacting the effectiveness of individuals and organizations, such as job satisfaction, organizational engagement, work commitment, work engagement, and turnover intention. These are the main factors impacting human resources management [50–52]. Most early studies of the relationship between work and family focused on conflicts be-

tween these two things. This research trend has shifted after some studies asserted that there was a special relationship between work and family [53]. According to Mark and MacDermid [54], work–life balance refers to the uniform allocation of interests and time to an individual's various roles; this is also called role balance. A person who dutifully fulfills all of their various roles, without neglecting any of them, can be said to have achieved role balance. Work–life balance was also defined as the perception of balance between work and family or other aspects of one's personal life [55]. According to Guest [56], work–life balance is a condition in which individuals can control when, where, and how to work and live by properly allocating their energy and time. Others defined it as a condition in which people have a feeling of satisfaction because they are equally engaged in all areas of their lives, without being overwhelmed in any area [57]. Another study defined work–life balance as balancing one's responsibilities to work and family [58]. Engagement in multiple roles can protect individuals from or reduce the impact of negative experiences in one role [59]. Hence, employees who have sufficient time and spatial resources to maintain a good work–life balance are able to commit more energy to their roles in their families. Employees who can fulfill their roles at work and at home have less tension and stress, which can arise from work–family conflicts, and they also experience work–life enhancement, which means that their performance in one domain is reinforced by that in another domain [53,60]. Employees in the hospitality industry sector suffer from unique emotional challenges due to the informal nature of this work and the high amount of emotional labor it requires. For those in today's complex workplace, work–life balance serves as the driving force for organizational life and is the ultimate goal [61,62].

### 2.3. Psychological Well-Being and Job Satisfaction

Some studies have examined the relationship between psychological well-being and job satisfaction. Deborah et al. [63] reported a close positive relationship between psychological well-being and job satisfaction. In a study on the effectiveness of happy workers, Wright and Cropanzano [64] argued that, of the factors studied, psychological well-being had the strongest positive impact on job satisfaction. Zeenat [65] argued that employee well-being had a considerable influence on job satisfaction, as it was linked to the well-being of an organization. Siu et al. [66] found that experiences that promoted psychological well-being at an organization could increase members' feelings of satisfaction. Muniandy [67] argued that psychological well-being had a significant positive impact on job satisfaction, and Capone et al. [68] reported that higher levels of psychological well-being were associated with higher levels of job satisfaction. Other studies have examined the positive impact of employees' psychological well-being on organizations or employees. Harris and Cameron [69] found that employee psychological well-being was closely related to engagement and identification with an organization. Kundi et al. [12] observed that promoting employees' psychological well-being could benefit organizations thanks to the positive impact of well-being on job-related attitudes and behaviors. Some studies have examined sub-factors of psychological well-being and job satisfaction. Jones et al. [16] reported that specific aspects of psychological well-being, including autonomy and opportunity, use of strengths, goal setting, and increased self-knowledge, could positively impact job satisfaction. Jung [70] reported that satisfaction had strong positive relationships with self-acceptance, environmental mastery, and purpose in life. Bansal et al. [71] found that one aspect of psychological well-being, purpose in life, had the strongest positive impact on employee job satisfaction. According to Slemp et al. [72], employees with high psychological well-being also had high levels of autonomy, which contributed to job satisfaction. Terry [73] observed that job satisfaction increased when autonomy was guaranteed. Clausen et al. [74] reported that increased autonomy was closely related to increased psychological well-being, which in turn increased job satisfaction. Xue et al. [75] said that employee well-being was one feature of a healthy working environment and concluded that the psychological well-being of organizational members played an important role in reducing turnover intention. Straume and Vitterso [76] argued that personal

growth was closely related to satisfaction. Ismail et al. [77] reported that well-being was strongly correlated with employees' happiness, joy, and personal growth, all of which enhanced satisfaction. Based on these results of previous studies, the following hypotheses are proposed:

**Hypothesis 1 (H1).** *Employees' psychological well-being positively influences job satisfaction.*

**Hypothesis 1a (H1a).** *Employees' self-acceptance positively influences job satisfaction.*

**Hypothesis 1b (H1b).** *Employees' positive relationships positively influence job satisfaction.*

**Hypothesis 1c (H1c).** *Employees' autonomy positively influences job satisfaction.*

**Hypothesis 1d (H1d).** *Employees' environmental mastery positively influences job satisfaction.*

**Hypothesis 1e (H1e).** *Employees' purpose in life positively influences job satisfaction.*

**Hypothesis 1f (H1f).** *Employees' personal growth positively influences job satisfaction.*

*2.4. Job Satisfaction and Pro-Social Service Behavior*

Several studies have investigated job satisfaction and pro-social behavior, including Lee [78], who found that job satisfaction was the most important factor predicting pro-social behavior, and more specifically, that satisfaction with one's salary made the largest contribution to promoting pro-social behavior. George and Brief [79] observed that, for employees in the service sector, job satisfaction had a strong positive impact on overall emotional state and that such emotions increased the likelihood of positive behaviors toward customers. Bettencourt and Brown [80] reported that employees' positive mental attitudes toward their jobs impacted pro-social behavior. MacKenzie et al. [81] observed that job satisfaction positively affected employees' pro-social service behavior because high job satisfaction promoted employee collaboration and excellent customer service. Asgari et al. [82] stated that employees with high job satisfaction were more likely to engage in pro-social behaviors, which were similar to organizational citizenship behavior. Jin et al. [83] found that hotel employees who were dissatisfied with their jobs did not engage in pro-social behavior. Therefore, the following hypothesis is proposed:

**Hypothesis 2 (H2).** *Employee job satisfaction positively influences pro-social service behavior.*

*2.5. Moderating Effect of Employee Work–Life Balance*

Some studies have investigated the moderating role of work–life balance. Kashyap et al. [84] argued that employees were organizational assets and that employee satisfaction and well-being were higher in organizations that supported good work–life balance. Yang et al. [85] reported that people who lacked a good work–life balance had poor psychological well-being, even after controlling for job-related and personal characteristics. According to Saraswati and Lie [86], employees who had positive emotions toward work tended to perform better, and this relationship was strengthened by a good work–life balance. Haar et al. [87] observed that a good work–life balance positively affected individual job and life satisfaction and negatively impacted anxiety and depression. French et al. [88] found that employees with good work–life balances were more satisfied with their jobs, performed better, and were more productive. Naithani [89] warned that organizations that overlooked work–life balance suffered from diminished work performance and declining productivity. Haider et al. [90] stated that work-life balance improved work performance while having a positive impact on psychological well-being, whereas Yang et al. [85] suggested that poor work-life balance could damage psychological well-being. Abdirahman et al. [91] argued that supporting work–life balance was one of the most important tasks of human resources management

and asserted that organizations should allow employees sufficient time to meet both family and work obligations. According to Aruldoss et al. [92], organizations should encourage pleasant working environments so that employees could maintain a good work–life balance, as this increased job satisfaction and engagement. Based on these findings, the following hypothesis is proposed (see Figure 1):

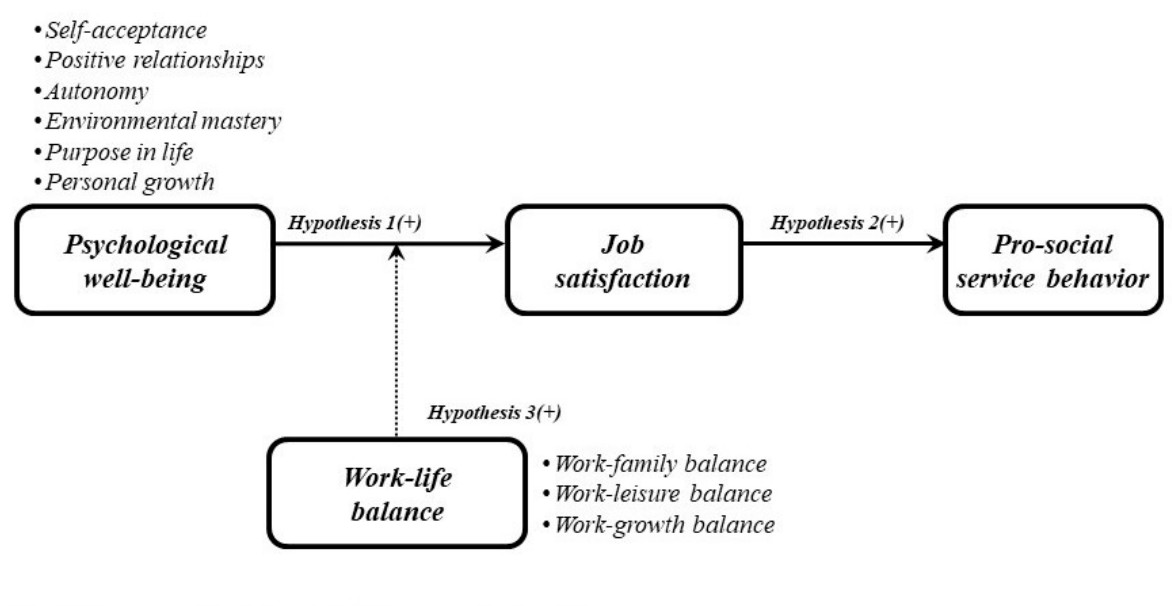

**Figure 1.** Research model.

**Hypothesis 3 (H3).** *The positive impact of employee psychological well-being on pro-social service behavior is strengthened by good work–life balance.*

## 3. Methodology

### 3.1. Sample and Data Collection

In this study, the sample comprised employees of deluxe hotels located in Seoul, South Korea. Data were collected via questionnaires that were first written in English by two researchers who were fluent in both English and Korean, then translated to Korean, and then re-translated back to English to ensure that no meaning was lost in translation [93]. Prior to the actual survey, a preliminary survey was conducted (50 copies to employees of hotel A in Seoul); the results of this data collection were used to revise the questionnaire. The pilot survey sample was excluded from the main survey. Data were collected from 1 September 2020 to 30 September 2020; the survey was distributed to employees at ten hotels. Out of a total of 22 deluxe hotels located in the Seoul area, only 10 hotels that allowed the survey were selected (Intercontinental Grand, Intercontinental Coex, Shilla, Lotte, Hyatt, W, Walkerhill, etc.). Participants were selected using convenience sampling. To ensure participant anonymity and data confidentiality, the returned responses were sealed and stored securely. A total of 400 questionnaires were distributed, and 275 completed surveys were included in the final analysis [94].

### 3.2. Instrument Development

The questionnaire used in this study included four sections. The first section measured the independent variable of psychological well-being, which was defined as a mindset where one's life had meaning and purpose according to one's own personal beliefs and

standards, as well as trying to lead a satisfactory life where the person positively perceived and accepted himself [35]. The psychological well-being scale (PWBS) developed by Ryff [35] was used to measure this variable. The scale included 18 questions measuring six factors of psychological well-being (self-acceptance, positive relationships, autonomy, environmental mastery, purpose in life, and personal growth); participants responded using a 7-point Likert scale to indicate agreement or disagreement with each statement. The second section of the survey measured the dependent variables of job satisfaction and pro-social service behavior. Job satisfaction was defined as an employee's mental attitude toward their job [95], and it was measured using five questions based on Spector [96]. Pro-social service behavior was defined as voluntary behavior by point-of-service employees that aimed to promote an organization's wealth by enhancing the quality of customer service [80]. This variable was measured using five questions based on Bettencourt and Brown [80]. The third section measured the moderating variable of work–life balance, which was defined as a sense of satisfaction in every domain of life due to appropriate allocation of personal resources [97]. Work–life balance was measured using 15 questions that divided this variable into three sub-factors: work–family balance, work–leisure balance, and work–growth balance. The final section collected data on respondents' demographic information (gender, age, education level, years of service, and department).

### 3.3. Data Analysis

The collected data were analyzed using the AMOS and SPSS statistics programs. An exploratory factor analysis using the Harmon test confirmed the absence of a CMV (common method variance) problem [98]. A confirmatory factor analysis, a reliability analysis, and CCR (composite construct reliability) and an AVE (average variance extracted) analysis were used to test the reliability and validity of the measurement items. A correlation analysis was used to determine whether the items correlated with the study's hypotheses, and the hypotheses were tested using a structural equation model and an MGA (multi-group analysis).

## 4. Results

### 4.1. Profile of the Sample

Of the respondents included in the final analysis, 69.1% were men, and 30.9% were women. A little over one-third (36.4%) of the respondents were in their twenties, 41.5% were in their thirties, and 22.2% were over 40. Most (80.0%) had a bachelor's degree. The largest cohort (43.6%) had worked from six to nine years in the service industry. Participants' departments included BOH (back of the house) (61.1%), FOH (front of the house) (31.6%), and other (7.3%). See Table 1.

**Table 1.** Profile of the sample (n = 275).

| Characteristic | N | Percentage |
|---|---|---|
| Gender | | |
| Male | 190 | 69.1 |
| Female | 85 | 30.9 |
| Age | | |
| 21 to 29 years | 100 | 36.4 |
| 30 to 39 years | 114 | 41.5 |
| Over 40 years | 61 | 22.2 |

**Table 1.** *Cont.*

| Characteristic | N | Percentage |
|---|---|---|
| Education level | | |
| University degree (4 years) | 220 | 80.0 |
| Grad. university degree (2 years) | 55 | 20.0 |
| Tenure | | |
| 5 years or fewer | 59 | 21.5 |
| 6–9 years | 120 | 43.6 |
| 10 years or more | 96 | 34.9 |
| Position | | |
| FOH (front of the house) | 87 | 31.6 |
| BOH (back of the house) | 168 | 61.1 |
| Other | 20 | 7.3 |

### 4.2. Measurement Model

Prior to the analysis, the existence of CMV was checked using Harman's one-factor test. The exploratory factor analysis showed that the explanatory power of the first single factor (27.618%) was less than half of the model's total explanatory power (72.968%). Furthermore, none of the measurement items explained most of the covariance, indicating an absence of severe bias [99]. These results confirmed that CMV was not a significant problem in the model. Table 2 shows the results of the validity and reliability tests.

**Table 2.** Confirmatory factor analysis and reliability analysis.

| Construct (Cronbach's Alpha) | Standardized Estimate | *t*-Value | CCR [a] | AVE [b] |
|---|---|---|---|---|
| Self-acceptance (0.784) | | | 0.817 | 0.555 |
| $PW_1$ | 0.827 | fixed | | |
| $PW_2$ | 0.734 | 11.535 *** | | |
| $PW_3$ | 0.668 | 11.559 *** | | |
| Positive relationships (0.768) | | | 0.779 | 0.537 |
| $PW_4$ | 0.727 | fixed | | |
| $PW_5$ | 0.776 | 10.661 *** | | |
| $PW_6$ | 0.680 | 9.784 *** | | |
| Autonomy (0.786) | | | 0.806 | 0.556 |
| $PW_7$ | 0.695 | fixed | | |
| $PW_8$ | 0.818 | 10.706 *** | | |
| $PW_9$ | 0.718 | 10.006 *** | | |
| Environmental mastery (0.784) | | | 0.830 | 0.558 |
| $PW_{10}$ | 0.704 | fixed | | |
| $PW_{11}$ | 0.850 | 11.496 *** | | |
| $PW_{12}$ | 0.677 | 9.909 *** | | |
| Purpose in life (0.868) | | | 0.867 | 0.690 |
| $PW_{13}$ | 0.806 | fixed | | |
| $PW_{14}$ | 0.843 | 14.923 *** | | |
| $PW_{15}$ | 0.843 | 14.924 *** | | |
| Personal growth (0.819) | | | 0.827 | 0.606 |
| $PW_{16}$ | 0.753 | fixed | | |
| $PW_{17}$ | 0.784 | 12.056 *** | | |
| $PW_{18}$ | 0.799 | 12.224 *** | | |

**Table 2.** *Cont.*

| Construct (Cronbach's Alpha) | Standardized Estimate | *t*-Value | CCR [a] | AVE [b] |
|---|---|---|---|---|
| Job satisfaction | | | 0.855 | 0.586 |
| (0.871) | | | | |
| $JS_1$ | 0.811 | fixed | | |
| $JS_2$ | 0.774 | 13.739 *** | | |
| $JS_3$ | 0.717 | 12.493 *** | | |
| $JS_4$ | 0.712 | 12.406 *** | | |
| $JS_5$ | 0.811 | 14.540 *** | | |
| Pro-social service behavior | | | 0.861 | 0.563 |
| (0.864) | | | | |
| $PSB_1$ | 0.700 | fixed | | |
| $PSB_2$ | 0.712 | 10.526 *** | | |
| $PSB_3$ | 0.786 | 11.468 *** | | |
| $PSB_4$ | 0.846 | 10.976 *** | | |
| $PSB_5$ | 0.698 | 10.343 *** | | |
| Work–life balance | | | 0.801 | 0.520 |
| (0.899) | | | | |
| Work–family balance | 0.629 | fixed | | |
| Work–leisure balance | 0.698 | 9.548 *** | | |
| Work–growth balance | 0.821 | 11.495 *** | | |
| Work–family balance | | | 0.829 | 0.603 |
| (0.875) | | | | |
| $WFB_1$ | 0.714 | fixed | | |
| $WFB_2$ | 0.648 | 10.201 *** | | |
| $WFB_3$ | 0.860 | 13.444 *** | | |
| $WFB_4$ | 0.883 | 13.737 *** | | |
| $WFB_5$ | 0.758 | 11.925 *** | | |
| Work–leisure balance | | | 0.861 | 0.648 |
| (0.901) | | | | |
| $WLB_1$ | 0.813 | fixed | | |
| $WLB_2$ | 0.861 | 16.358 *** | | |
| $WLB_3$ | 0.790 | 14.594 *** | | |
| $WLB_4$ | 0.770 | 14.102 *** | | |
| $WLB_5$ | 0.792 | 14.629 *** | | |
| Work–growth balance | | | 0.889 | 0.672 |
| (0.908) | | | | |
| $WGB_1$ | 0.732 | fixed | | |
| $WGB_2$ | 0.826 | 13.578 *** | | |
| $WGB_3$ | 0.830 | 13.645 *** | | |
| $WGB_4$ | 0.865 | 14.238 *** | | |
| $WGB_5$ | 0.841 | 13.832 *** | | |

Note: [a] CCR = composite construct reliability; [b] AVE = average variance extracted; standardized estimate = β-value; $\chi^2$ = 1625.512 (df = 821) $p < 0.001$; $\chi^2/df$ = 1.980; goodness-of-fit index (GFI) = 0.800; Tucker–Lewis Index (TLI) = 0.865; comparative fit index (CFI) = 0.877; incremental fit index (IFI) = 0.879; root square error of approximation (RMSEA) = 0.060; *** $p < 0.001$.

A secondary confirmative factor analysis was then conducted. All measurement items had a statistically significant impact ($p < 0.001$), with a Cronbach's alpha and CCR over 0.8 and an AVE over 0.5, which satisfied the criteria [100]. Furthermore, the AVEs for each measurement item were larger than the squares of the coefficients, which confirmed the validity of the measurement items (Table 3). The model had satisfactory goodness of fit ($\chi^2$ = 1625.512; $\chi^2/df$ = 1.980; GFI = 0.800; TLI = 0.865; CFI = 0.877; IFI = 0.879; RMSEA = 0.060). Table 3 shows the results of the correlation analysis of the measurement items, which indicate that the correlation coefficients were consistent with the direction of the hypotheses [101].

**Table 3.** Means, standard deviations, and correlation analyses.

| Construct | 1 | 2 | 3 | 4 | 5 | 6 | 7 | 8 | Mean $\pm$ SD [a] |
|---|---|---|---|---|---|---|---|---|---|
| 1. Self-acceptance | 1 | | | | | | | | $5.26 \pm 0.77$ |
| 2. Positive relationships | 0.493 ** | 1 | | | | | | | $5.38 \pm 0.80$ |
| 3. Autonomy | 0.362 ** | 0.272 ** | 1 | | | | | | $5.10 \pm 0.78$ |
| 4. Environmental mastery | 0.413 ** | 0.354 ** | 0.457 ** | 1 | | | | | $5.21 \pm 0.73$ |
| 5. Purpose in life | 0.386 ** | 0.331 ** | 0.487 ** | 0.484 ** | 1 | | | | $5.24 \pm 0.89$ |
| 6. Personal growth | 0.425 ** | 0.362 ** | 0.401 ** | 0.515 ** | 0.517 ** | 1 | | | $5.26 \pm 0.84$ |
| 7. Job satisfaction | 0.380 ** | 0.336 ** | 0.319 ** | 0.320 ** | 0.334 ** | 0.414 ** | 1 | | $5.14 \pm 0.88$ |
| 8. PSB | 0.398 ** | 0.470 ** | 0.253 ** | 0.351 ** | 0.352 ** | 0.303 ** | 0.342 ** | 1 | $5.25 \pm 0.77$ |
| 9. Work–life balance | 0.295 ** | 0.307 ** | 0.308 ** | 0.388 ** | 0.340 ** | 0.384 ** | 0.377 ** | 0.315 ** | $5.05 \pm 0.76$ |

Note: [a] SD = standard deviation. All variables were measured on a 7-point Likert scale from 1 (strongly disagree) to 7 (strongly agree); ** $p < 0.01$.

### 4.3. Structural Equation Modeling

Structural equation modeling (SEM) was used to test the hypotheses; the results are shown in Table 4. The final structural model had appropriate goodness of fit ($\chi^2 = 880.865$; $\chi^2/df = 2.606$; GFI = 0.803; IFI = 0.853; CFI = 0.810; RMSEA = 0.077). Hypothesis 1 was tested to verify the impact of deluxe hotel employees' psychological well-being on job satisfaction; the results were as follows. Of the sub-factors of psychological well-being, positive relationships ($\beta = 0.471$; t = 5.757; $p < 0.001$), self-acceptance ($\beta = 0.193$; t = 2.236; $p < 0.05$), purpose in life ($\beta = 0.169$; t = 2.602; $p < 0.01$), and environmental mastery ($\beta = 0.161$; t = 2.238; $p < 0.05$) positively impacted job satisfaction. However, autonomy ($\beta = -0.015$; t = $-0.176$; $p > 0.05$) and personal growth ($\beta = -0.046$; t = $-0.533$; $p > 0.05$) did not have significant effects. Therefore, Hypothesis 1 was only partially accepted. These findings are consistent with those of Jones et al. [16] and Jung [70] and imply that people with high levels of psychological well-being were more likely to be satisfied with their jobs. Hypothesis 2, which examined the impact of employees' job satisfaction on pro-social service behavior, was accepted, as job satisfaction ($\beta = 0.431$; t = 5.750; $p < 0.001$) positively impacted pro-social service behavior. This finding aligns with those of Bettencourt and Brown [80] and Jin et al. [83], who also found that people who were satisfied with their jobs were more likely to engage in pro-social behavior.

**Table 4.** Structural parameter estimates.

| Hypothesized Path (Stated as Alternative Hypothesis) | Standardized Path Coefficients | *t*-Value | Results |
|---|---|---|---|
| H1a: Self-acceptance → Job satisfaction | 0.193 | 2.236 * | Accepted |
| H1b: Positive relationships → Job satisfaction | 0.471 | 5.757 *** | Accepted |
| H1c: Autonomy → Job satisfaction | −0.005 | −0.059 | Rejected |
| H1d: Environmental mastery → Job satisfaction | 0.161 | 2.238 * | Accepted |
| H1e: Purpose in life → Job satisfaction | 0.169 | 2.602 ** | Accepted |
| H1f: Personal growth → Job satisfaction | −0.021 | −0.247 | Rejected |
| H2: Job satisfaction → Pro-social service behavior | 0.431 | 5.750 *** | |
| Goodness-of-fit statistics | $\chi^2_{(338)} = 880.865$ ($p < 0.001$) | | |
| | $\chi^2/df = 2.606$ | | |
| | GFI = 0.803 | | |
| | IFI = 0.853 | | |
| | CFI = 0.810 | | |
| | RMSEA = 0.077 | | |

Note: * $p < 0.05$, ** $p < 0.01$ and *** $p < 0.001$; GFI = goodness-of-fit index; NFI = normed fit index; CFI = comparative fit index; RMSEA = root mean square error of approximation.

### 4.4. Moderating Effects of Work–Life Balance

According to Hypothesis 3, the impact of deluxe hotel employees' psychological well-being on job satisfaction was moderated by employee work–life balance. To analyze

this moderating effect, the subjects were divided into two groups based on their average reported level of work–life balance. The degree of freedom in the constrained model was compared to that in the unconstrained model to measure the significance of the moderating effect. This analysis (Table 5) showed that the positive effect of psychological well-being on job satisfaction differed depending on an employee's work–life balance, and, hence, Hypothesis 3 was partially accepted. Of the variables of psychological well-being, the positive impact of purpose in life on pro-social service behavior was particularly strong among participants with high work–growth balance ($\beta = 0.334$; t = 4.438; $p < 0.001$) compared to those with low work–growth balance ($\beta = -0.090$; t = $-1.209$; $p > 0.05$). Therefore, only Hypothesis 3e was accepted. This result can be interpreted to indicate that, among people with high levels of work–life balance, individual beliefs about the meaning, goal, and direction of one's life tended to have a greater impact on job satisfaction. Because a high level of work–life balance positively affected the meaningfulness or performance of a job, it also increased job satisfaction [87,88].

**Table 5.** Moderating effects of employee work–life balance.

| Hypothesized Path | Low WLB (N = 177) | | High WLB (N = 223) | | Constrained Model $\chi^2$ (df = 439) | $\Delta\chi^2$ (df = 1) |
|---|---|---|---|---|---|---|
| | Standardized Coefficients | *t*-Value | Standardized Coefficients | *t*-Value | | |
| H3a: Self-acceptance → Job satisfaction | 0.218 | 2.100 * | 0.198 | 1.478 | 1759.643 | 0.038 |
| H3b: Positive relationships → Job satisfaction | 0.498 | 5.331 *** | 0.428 | 4.922 *** | 1759.624 | 0.019 |
| H3c: Autonomy → Job satisfaction | −0.025 | −0.199 | 0.004 | 0.042 | 1759.637 | 0.032 |
| H3d: Environmental mastery → Job satisfaction | 0.180 | 1.817 | 0.088 | 0.887 | 1757.787 | 0.182 |
| H3e: Purpose in life → Job satisfaction | −0.090 | −1.209 | 0.334 | 4.438 *** | 1767.941 | 8.336 * |
| H3f: Personal growth → Job satisfaction | 0.127 | 1.069 | −0.146 | −1.084 | 1761.509 | 1.904 |

Note: unconstrained model $\chi^2$ = 1759.605; df = 438; CFI = 0.741; AIC (Akaike information criterion) = 1987.064; * $p < 0.05$ and *** $p < 0.001$.

## 5. Discussion and Implications

### 5.1. Discussion of Results

This study established a framework for understanding the organic, causal relationships among deluxe hotel employees' psychological well-being, job satisfaction, and pro-social service behavior. It also explored the moderating role of work–life balance in the relationship between psychological well-being and job satisfaction. First, of the studied sub-factors of employee psychological well-being, positive relationships increased job satisfaction the most, followed by self-acceptance, purpose in life, and environmental mastery [70]. Second, deluxe hotel employees' job satisfaction positively impacted their pro-social service behavior; in other words, employees who were more satisfied with their jobs were more likely to voluntarily engage in behaviors that improved customer service [80]. Third, the positive effect of one sub-factor of psychological well-being, purpose in life, had a stronger impact on job satisfaction in respondents with high levels of work–life balance. This result implied that, when people believe that their work and life are well-balanced, their perception of meaning in their lives has a stronger impact on their job satisfaction [88].

### 5.2. Theoretical and Practical Implications

The findings of this study have the following theoretical implications. First, following current research trends, this study examined the impact of psychological well-being on employees in the hospitality industry and demonstrated that job satisfaction and pro-social service behavior could differ among members of organizations in the hospitality industry. Very few studies so far have examined this organic, causal relationship among employees of deluxe hotels. Therefore, the present study provides a theoretical basis linking psychological well-being to job satisfaction and behavior. The value of this study also lies in its role as a pioneering study focused on employees of deluxe hotels. More specifically, the positive impact of psychological well-being on organizations was demonstrated using

specific performance variables. The findings of this study may contribute to the development of plans to promote psychological well-being among hospitality employees. Another theoretical implication is that this study verified the moderating role of work–life balance in the relationships of psychological well-being with job satisfaction and behavior. Today, balancing work and one's personal life has become more important than ever as more and more young people are entering the hospitality industry. Therefore, the present study investigated an influential moderating variable. This finding is particularly significant as the present study is the first to explore this relationship. Furthermore, this study examined the impact of employee psychological well-being on voluntary customer-service behavior and empirically demonstrated that happy people (defined as those with high levels of psychological well-being) more frequently offered to help others at work and in their private lives. This study showed that perceived psychological well-being increased job satisfaction, which further enhanced pro-social service behavior in general.

This study also has several practical implications. First, it can help organizations understand the effects of psychological well-being on employees' attitudes and behaviors. This study confirmed that the perception of psychological well-being positively impacted deluxe hotel employees' pro-social service behavior via job satisfaction. In line with these findings, organizations should work to boost employees' psychological well-being; this may include efforts to improve employees' mental health. Promoting employees' psychological well-being can enable managers to encourage staff development and may increase employee commitment to an organization, leading to improved work performance. This implies that organizations play a critical role in enhancing employees' psychological well-being, as this factor not only improved employee performance, but also increased job satisfaction and behaviors aimed at helping customers. In other words, managers should pay close attention to employees' psychological well-being in order to increase voluntary customer-focused behavior. Hence, organizations should create constructive systems that can improve employees' psychological well-being. In particular, it is necessary to let employees know that an organization is keenly interested in promoting their psychological well-being. Since well-being based on positive relationships with others had the largest impact on work performance, organizations should encourage informal employee gatherings for leisure or sports. It is also necessary to present guidelines or directions for employees to gain encouragement and empathy through close, cooperative relationships with their colleagues and superiors in an organization. In addition, these smooth, interpersonal relationships enable them to share new ideas, perspectives, and experiences that they cannot create on their own, which can become a source of positive attitudes and behaviors within an organization. Coaching is also important; therefore, organizations should offer and encourage a range of workshops through collaborations with educational institutions or sports clubs. Developing innovative ways to regularly check employees' well-being, such as via surveys or smooth communication, may be helpful as well. Employee voluntary service behavior is particularly important. Such behavior may involve providing information in response to customer requests, regardless of a staff member's role and responsibilities, and these behaviors often help organizations secure a competitive advantage and distinguish themselves from other companies. Organizations should also encourage members to come up with ways to improve service provision [102]. Second, this study demonstrated that the positive impact of psychological well-being on job satisfaction was strengthened by a good work–life balance. Even if organizational supports for psychological well-being are insufficient, the positive impact of psychological well-being on job performance and satisfaction can be enhanced by good work–life balance. Therefore, efforts should be made at the organizational level to support employees' work–life balance and their perceptions of this balance. For example, organizations can provide assistance, such as counseling services, for staff members and their families. They can also develop supportive systems that help workers cope with work pressure and time management so that they do not feel pressured or obliged to work extra hours. It is also important to ensure that work does not intrude on employees' personal lives, either physically or temporally. Psychological well-being had a

stronger positive impact when employees felt that an organization respected their need to balance work with their private life.

### 5.3. Limitations and Future Research

This study also has several limitations. First, because the participants were limited to employees at deluxe hotels in Korea, it is difficult to generalize the results to other Korean industries or to other branches of the hospitality industry. In addition, another limitation is that the survey was performed two years ago. Future studies should explore these relationships in other cultures, industries, and/or branches. Second, since the data used in this study were self-reported, it is possible that the participants gave responses that they considered desirable or socially acceptable. Although the possibility of CMB error was adequately tested, future studies should seek to address this shortcoming. Third, this study only included one moderating variable: work–life balance. Future studies should examine the impacts of a range of possible moderating variables on the relationship between employees' psychological well-being and job satisfaction (personal and job-related variables, etc.). Finally, although pro-social service behavior was used as the final dependent variable in this study, future studies should also examine the impacts of psychological well-being on other behavioral variables, such as turnover intention.

**Author Contributions:** Data curation, Y.-H.H. and H.-S.J.; Formal analysis, Y.-H.H. and H.-S.J.; Investigation, Y.-H.H., H.-S.J. and H.-H.Y.; Methodology, Y.-H.H.; Resources, H.-H.Y.; Software, H.-S.J.; Supervision, H.-H.Y.; Validation, H.-S.J. and H.-H.Y.; Visualization, H.-H.Y.; Writing—original draft, Y.-H.H. and H.-S.J. All authors have read and agreed to the published version of the manuscript.

**Funding:** This research received no external funding.

**Institutional Review Board Statement:** Not applicable.

**Informed Consent Statement:** Not applicable.

**Data Availability Statement:** The data presented in this study are available on request from the first author.

**Conflicts of Interest:** The authors declare no conflict of interest.

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
