# Peer review of "Impact of Hotel Employees’ Psychological Well-Being on Job Satisfaction and Pro-Social Service Behavior: Moderating Effect of Work–Life Balance"

_sustainability, doi:10.3390/su151511687_

Round 1
Reviewer 1 Report
Thank you for allowing me to review this paper. However, I have shared my viewpoints.
In the abstract part, you must include a short methodology (sample size, sampling technique). Besides, you must include the implications, limitations and areas for future research.
The introduction part is well-written. But, you need to give a little touch on any theory. For example, you have used Work-life Balance and Job satisfaction; in that case, you can use the Conservation of Recourses Theory (COR).
The literature review part needs to improve by adding theoretical support, as you formulated some hypotheses. Your variables must come from theory; then, you can formulate the hypotheses. You are requested to read and improve your literature review by adding the concept of the fowling paper.
https://www.emerald.com/insight/content/doi/10.1108/JEC-05-2020-0098/full/html
The methodology section is well structured. You have written, “Participants were selected using convenience sampling”. But, you need to give a justification with reference. Besides, you have written, “275 completed surveys were included in the final analysis”…You need to prove that, the sample size is adequate. However, you will get a clear idea about the raising issue from the following article.
Sampling Techniques (Non-Probability) and sample size determination
http://scientificia.com/index.php/JEBE/article/view/201
In the result section, it is quite impressive except Table No 5 (Moderating Effect). You have used Multi-group Analysis (MGA); you can do that but requested to check the result in Table 5.
The discussion and contribution parts must be separated.
One of the main shortcomings of this article is not to discuss the theoretical contributions.
Finally, you must include “Conclusion, limitations, and areas for future research.”
Thanks.
Author Response
Reviewer 1
→ We greatly appreciate the time taken by the reviewer to point out the specific areas where improvements were needs. We sincerely hope that you remain safe from the coronavirus.
Comment#1: In the abstract part, you must include a short methodology (sample size, sampling technique). Besides, you must include the implications, limitations and areas for future research. → Thank you for your comments. We revised it.
Comment#2: The introduction part is well-written. But, you need to give a little touch on any theory. For example, you have used Work-life Balance and Job satisfaction; in that case, you can use the Conservation of Recourses Theory (COR). → We thank the reviewer for this comment. The COR of resources is slightly different from what this study is trying to clarify, as some critical resources affect stress. In the introduction, we supplemented the clearer research purpose and background. We appreciate the time and effort you put into reviewing our manuscript.
Comment#3: The literature review part needs to improve by adding theoretical support, as you formulated some hypotheses. Your variables must come from theory; then, you can formulate the hypotheses. You are requested to read and improve your literature review by adding the concept of the fowling paper. → We appreciate the reviewer’s insight. After defining variables, previous studies related to hypotheses were mentioned. Thank you for your valuable review.
Comment#4: The methodology section is well structured. You have written, “Participants were selected using convenience sampling”. But, you need to give a justification with reference. Besides, you have written, “275 completed surveys were included in the final analysis”…You need to prove that, the sample size is adequate. However, you will get a clear idea about the raising issue from the following article. . → Thank you for your comments. We revised it.
Comment#5: In the result section, it is quite impressive except Table No 5 (Moderating Effect). You have used Multi-group Analysis (MGA); you can do that but requested to check the result in Table 5. → Thank you for your comments. The explanation has been further supplemented.
Comment#6: The discussion and contribution parts must be separated. One of the main shortcomings of this article is not to discuss the theoretical contributions. Finally, you must include “Conclusion, limitations, and areas for future research. → We thank the reviewer for this comment. 5.1, 5.2, and 5.3 sub-numbers were entered and classified, and theoretical contributions were supplemented.
Reviewer 2 Report
Review: The impact of hotel employees' psychological well-being on job satisfaction and pro-social service behavior: moderating effect of work-life balance
STUDY
The study investigates the effect that balance between work and private life affect job satisfaction and promotes pro-social behaviour. The study has a unique approach in that it looked mainly at luxurious hotels that mainly served high income visitors.
Abstract
· The Abstract seems to be missing some elements necessary to promote the study:
o The first few lines do state what the problem is and why the study is being done.
o The authors should also include a short sentence or two on methodology and the results and why the study was important.
Keywords
· Keywords are sufficient in number and relevance
Introduction & Lit Review
· The introduction is sufficient and the Lit Review contains the necessary information and past studies.
Methods.
· Sentence 257 to 265 needs some attention. It is assumed that 50 participants took part in a pilot study? This needs to be made clear.
· The inclusion and exclusion criteria for hotels selected is not stated as well as why 400 participants were invited. How did the authors decide on the number of participants?
· Reasons as to the large fall-out? (275)
· It seems that the authors designed their own questionnaire? What is the reliability and validity, norming, etc?
· It is unclear whether this study was approved by an Ethics Committee. Just before the References, it states that “Institutional Review Board Statement” – Not applicable. Results
· Good use of tables and graphics
· I would prefer that a statistician confirm the results
Discussion
· The Discussion needs to integrate Lit review and Results.
· Also, do not refer to the results in Table…, but in a short sentence report on the results without repeating the Results. It helps the reader not to go forwards and back.
Conclusion
· This is a very important study and deserves to be published.
References
· Some very old resources used and a few current ones.
Strengths of the paper
· Very important topic
· Seemingly good and robust analysis
Weakness of the paper
· No Institutional ethics clearance
· Quality of Presentation: Good
· Interest to the Readers: Readers will be very interested in this study and will appeal to a wide audience.
· Overall Merit: This study is very interesting and can have merit in being published
· English Level: Good.
Author Response
Comments to the Author: The study investigates the effect that balance between work and private life affect job satisfaction and promotes pro-social behaviour. The study has a unique approach in that it looked mainly at luxurious hotels that mainly served high income visitors.→ We greatly appreciate the time taken by the reviewer to point out the specific areas where improvements were needs. We sincerely hope that you remain safe from the coronavirus.
Comment#1: The Abstract seems to be missing some elements necessary to promote the study: The first few lines do state what the problem is and why the study is being done. The authors should also include a short sentence or two on methodology and the results and why the study was important. Keywords are sufficient in number and relevance→ Thank you for your comments. We revised it.
Comment#2: The introduction is sufficient and the Lit Review contains the necessary information and past studies. → We thank the reviewer for this comment.
Comment#3: Sentence 257 to 265 needs some attention. It is assumed that 50 participants took part in a pilot study? This needs to be made clear. The inclusion and exclusion criteria for hotels selected is not stated as well as why 400 participants were invited. How did the authors decide on the number of participants? Reasons as to the large fall-out? (275) It seems that the authors designed their own questionnaire? What is the reliability and validity, norming, etc? It is unclear whether this study was approved by an Ethics Committee. Just before the References, it states that “Institutional Review Board Statement” – Not applicable. Results. → We appreciate the reviewer’s insight. We revised it. Pre-study samples were not used in the main study. The size of the sample was determined by referring to previous studies and mentioned in the text. The questionnaire items were extracted through previous research, and the author did not arbitrarily modify the items. All of these contents were mentioned and revised in the text. Also, this papers that meet the requirements for exemption from approval by the IRB.
Comment#4: Th The Discussion needs to integrate Lit review and Results. Also, do not refer to the results in Table…, but in a short sentence report on the results without repeating the Results. It helps the reader not to go forwards and back.
. . → We thank the reviewer for this comment. 5.1, 5.2, and 5.3 sub-numbers were entered and classified, and theoretical contributions were supplemented.
Comment#5: This is a very important study and deserves to be published.
→ Thank you for your comments.
Round 2
Reviewer 1 Report
Thank you for the overall improvement of your manuscript.
However, still, you need to improve the discussion part; you are requested to discuss each of your findings in relation to other researchers' findings. You will get the idea regarding the discussion part from the following article.
https://www.emerald.com/insight/content/doi/10.1108/JEC-05-2020-0098/full/html
Best of luck.
Author Response
Thank you for your careful review. The discussion section has been developed more abundantly by citing previous studies.